# Computation of the Deuteron Mass and Force Unification via the Rotating Lepton Model

**Constantinos G. Vayenas** [1,2,*] **, Dimitrios Grigoriou** [2] **, Dionysios Tsousis** [2] **, Konstantinos Parisis** [3]
**and Elias C. Aifantis** [3,4,5,*]

1   Academy of Athens, Panepistimiou 28 Ave., GR-10679 Athens, Greece
2   Department of Chemical Engineering, University of Patras, Caratheodory 1 St, GR-26504 Patras, Greece
3   School of Engineering, Aristotle University of Thessaloniki, GR-54124 Thessaloniki, Greece
4   College of Engineering, Michigan Technological University, Houghton, MI 49931, USA
5   Mechanical Engineering, Friedrich-Alexander University Erlangen-Nuremberg, 90762 Fürth, Germany
*   Correspondence: cgvayenas@upatras.gr (C.G.V.); mom@mom.edu (E.C.A.)

**Abstract:** The rotating lepton model (RLM), which is a 2D Bohr-type model of three gravitating rotating neutrinos, combining Newton's gravitational law, special relativity, and the de Broglie equation of quantum mechanics, and which has already been used to model successfully quarks and the strong force in several hadrons, has been extended to 3D and to six rotating neutrinos located at the vertices of a normal triangular octahedron in order to compute the Lorentz factors, gamma, of the six neutrinos and, thus, to compute the total energy and mass of the deuteron, which is the lightest nucleus. The computation includes no adjustable parameters, and the computed deuteron mass agrees within 0.05% with the experimental mass value. This very good agreement suggests that, similarly to the strong force in hadrons, the nuclear force in nuclei can also be modeled as relativistic gravity. This implies that, via the combination of special relativity and quantum mechanics, the Newtonian gravity gets unified with the strong force, including the residual strong force.

**Keywords:** nuclear force; residual strong force; deuteron interaction energy; special relativity; gravitational mass; de Broglie wavelength; rotating lepton model

**MSC:** 70-XX; 83-XX

## 1. Introduction

The strong force or strong interaction binds together the constituents of protons, neutrons, and other hadrons, such as quarks and gluons. It is also the force holding together protons and neutrons to form nuclei. In this case, the strong force is known as nuclear force or the residual strong force. The strong force is thought to be mediated by gluons acting upon quarks, antiquarks and gluons themselves as elaborated upon in the theory of quantum chromodynamics (QCD) [1]. Using special relativity [2,3], in conjunction with the de Broglie equation of quantum mechanics, recent work has shown that quarks have highly relativistic rotational velocities, and that their rest mass is that of neutrinos [4–6]. These particles rotate with speeds close to the speed of light; thus, their inertial and gravitational masses become enormous [7], reaching the range of the Planck mass. This suggests that the strong force or strong interaction can be viewed as gravitational attraction between ultrarelativistic quarks or neutrinos [4–6,8–10]. This is the basis of the rotating lepton model (RLM), which is a two-dimensional (2D) Bohr-type model with gravity as the attractive force [4,5,11], based entirely on three basic equations of Newton, Einstein, and de Broglie, thus combining special relativity and quantum mechanics and leading to semiquantitative (±1%) agreement with the measured hadron and boson masses [11]. Here, the RLM is extended to 3D and is used for the first time to compute the mass of a nucleus, i.e., of the deuteron. The computed mass agrees within 0.05% with the experimental deuteron mass without any adjustable parameters. This

very good agreement shows that, similarly to the strong force [11,12], the nuclear force can also be modeled as relativistic gravity.

Gravitational forces between neutrinos or within neutrino–electron pairs are commonly expected to be very weak. This is because the rest masses of neutrinos are very small (Figure 1). However, special relativity [2,3] in conjunction with the equivalence principle [7] dictates that the gravitational masses of neutrinos [4,5] and, thus, the gravitational forces between neutrinos and electrons tend to infinity as their speeds approach the speed of light [4,5,13]. Accordingly, due to their very small rest masses, neutrinos can be easily accelerated to ultrarelativistic speeds and be caught in circular rotational states with speeds very close to the speed of light [4,5]. It has, thus, been shown that these ultrarelativistic rotating neutrinos have the properties of quarks [4–11], i.e., their relativistic mass is equal to the rest mass of quarks. This has led to the development of the rotating lepton model (RLM), which allows for the computation of the masses of hadrons and bosons [4–10] with an astonishing precision of 1% without any adjustable parameters.

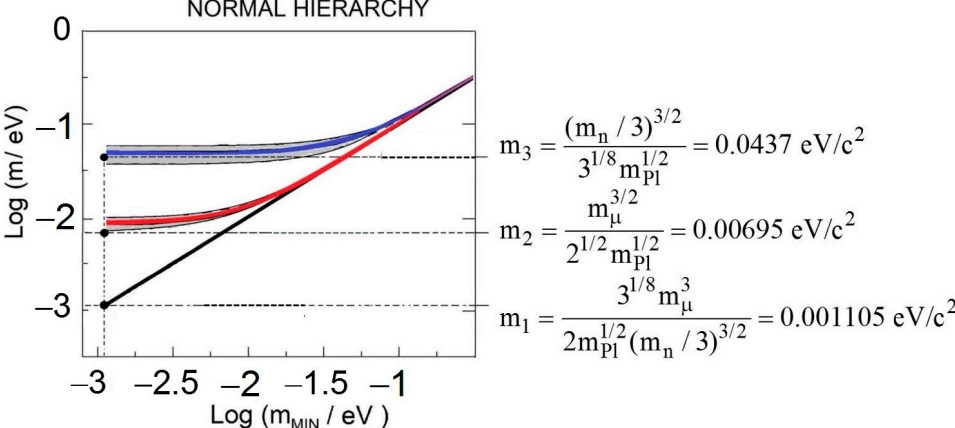

**Figure 1.** Comparison of computed via the RLM (horizontal dotted lines) [12] and experimental [14,15] neutrino eigenmasses; $m_n$ and $m_\mu$ are the neutron and muon masses, respectively. The $m_3$, $m_2$, and $m_1$ expressions were retrieved from [12]. These equations imply that, interestingly, $m_3/m_2 = m_2/m_1 \approx 6.3 \approx 2\pi$.

## 2. The Rotating Lepton Model (RLM)

An example for the formation of a neutron from three ultrarelativistic gravitationally confined rotating neutrinos acting as quarks is shown in Figure 2. The equation of motion for each rotating neutrino is obtained by combining the special relativistic equation for circular motion for each rotating neutrino,

$$F_G = \gamma m_o v^2/r, \tag{1}$$

where $m_o$ is the neutrino rest mass, and $\gamma$ is the Lorentz factor $(1 - v^2/c^2)^{-1/2}$, with Newton's gravitational law using the gravitational masses of the neutrinos. According to special relativity [2–4], $\gamma m_o$ is equal to the relativistic mass of a particle of rest mass $m_o$, while $\gamma^3 m_o$ is equal to its inertial mass [2–4]. According to the equivalence principle, the latter is equal to the gravitational mass, $m_g$, i.e.,

$$m_g = \gamma^3 m_o. \tag{2}$$

It is worth noting that the gravitational mass, $m_g$, is defined from Newton's gravitational law, i.e., from

$$m_g^2 = \sqrt{3}F_G r^2/G, \tag{3}$$

thus satisfying by definition Equation (4).

$$F_G = (Gm_g^2)/(\sqrt{3}r^2). \tag{4}$$

Thus, upon combining with Equations (1) and (2), one obtains

$$\frac{\gamma m_o v^2}{r} = \frac{G m_g^2}{\sqrt{3}r^2} = \frac{G m_o^2 \gamma^6}{\sqrt{3}r^2}. \tag{5}$$

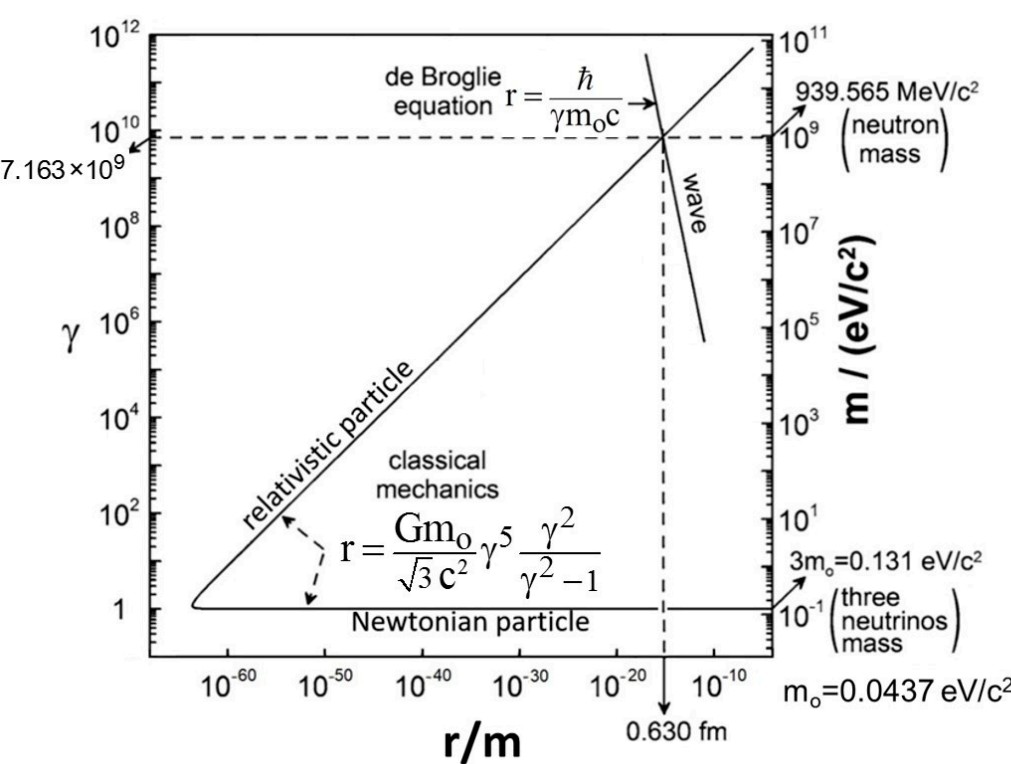

**Figure 2.** Plot of the relativistic equation for circular motion of three gravitating neutrinos each of rest mass $m_o$ (Equation (6)) and of the de Broglie equation for each of these particles (Equation (7)), which accounts for its dual wave–particle nature, showing that the intersection of these two curves defines the neutron mass (939.565 MeV/c$^2$) and radius (0.630 fm) in quantitative agreement with experiment [16]. The heaviest neutrino mass value used ($m_o$ = 0.0437·eV/c$^2$) is computed from the equation $m_o = m_n^{3/2}/3^{1/8}m_{Pl}^{1/2}$ derived from Equations (5) and (7), and it lies within the current experimental uncertainty limits of the heaviest neutrino mass measured at the Super-Kamiokande [14,15]. The neutron mass (939.565 MeV/c$^2$) is a factor of $\gamma$ (=7.163 $\times$ 10$^9$) larger than the rest mass of the three neutrinos (0.131 eV/c$^2$) according to Equation (9).

Thus, Equation (5) becomes

$$r = \left(\frac{G m_o}{\sqrt{3}c^2}\right)\gamma^5\left[\gamma^2/(\gamma^2-1)\right]. \tag{6}$$

This equation is plotted in Figure 2 for $m_o$ = 0.0437 eV/c$^2$ [4–10], i.e., a value within the range of the heaviest neutrino mass, 0.048 $\pm$ 0.01 eV/c$^2$ [4,14,15], and exhibits bistability, i.e., there are two $\gamma$ values satisfying Equation (6) for any r value above $(2.96)G m_o/c^2(= 1.71 \times 10^{-64}$ m) [4]. The high $\gamma$ branch corresponds to highly relativistic speeds; as shown in Figure 2, the model solution is found by the intersection of Equation (6) with the de Broglie equation

$$r = \hbar/\gamma m_o v = \hbar/mv, \tag{7}$$

which reduces for $\gamma > 10^2$ to

$$r = \hbar/\gamma m_o c = \hbar/mc, \tag{8}$$

where $m(= \gamma m_o)$ is the relativistic neutrino mass. The intersection of Equations (6) and (8) defines the following $\gamma$ and neutron mass, $m_n$, values:

$$\gamma = 7.163 \times 10^9; \; m_n = 3\gamma m_o = 939.565 \text{ MeV}/c^2, \tag{9}$$

as shown in Figure 2. The computed neutron mass, $m_n$, is in quantitative agreement with the literature value [1]. Furthermore, the r value at the intersection of Figure 2 is also computed from the de Broglie or Compton equation:

$$r = \frac{3\hbar}{m_n c} = 0.630 \text{ fm}, \tag{10}$$

which is in quantitative agreement with the value of the proton radius measured recently by Burkert and coworkers [16], via deep inelastic Compton scattering, i.e., 0.63 fm.

Equations (6) and (8) also yield the following expressions, relating $m_n$ and $m_o$:

$$m_n = 3^{13/12}(m_{Pl}m_o^2)^{1/3}; \; m_o = (m_n/3)^{3/2}/(3^{1/8}m_{Pl}^{1/2}), \tag{11}$$

i.e., $m_n$ = 939.565 MeV/$c^2$ [13] and $m_o = 0.0437250$ eV/$c^2$ [4,11,12], where the $m_o$ value lies again within the range of the heaviest neutrino mass, $(0.048 \pm 0.01$ eV/$c^2)$ [4,11,12], and where $m_{Pl}(= \hbar c/G)^{1/2} = 1.220890 \times 10^{28}$ eV/$c^2$.

These studies [4–11,13] show strongly that the strong force can be modeled successfully as relativistic gravity between relativistic neutrinos or, equivalently, between quarks, and that the weak force can be modeled successfully as relativistic gravity between neutrinos and electrons/positrons [6,8,11,13].

Here, we show that gravity, special relativity, and the RLM suffice to describe not only the strong force, i.e., the intrahadron forces between quarks, but also the residual strong forces, i.e., the interhadron forces, commonly termed nuclear forces, between hadrons in nuclei [1,4]. We are, thus, able to model the structure, as well as compute the mass and interaction energy of the deuteron, formed from a proton and a neutron, within 0.05%.

### 3. The Deuteron Model

The deuteron, i.e., the nucleus of the deuterium, consists of a proton and a neutron and has a mass of 1875.613 MeV/$c^2$, which is 2.225 MeV/$c^2$ smaller than the sum of the proton (938.272 MeV/$c^2$) and neutron (939.565 MeV/$c^2$) masses, respectively. The difference is commonly called the interaction energy.

According to the rotating lepton model (RLM), the neutron, as well as the proton, consists of three neutrinos of the heaviest rest mass, $m_o$, eigenstate, i.e., of mass 3 in the normal hierarchy [4,5,14,15] (Figure 1). These neutrinos in their gravitationally confined relativistic state have $\gamma$ values as high as $10^{10}$; hence, their relativistic mass is of the order of 0.3 GeV/$c^2$, i.e., in the range of quarks, thus acting as u and/or d quarks. Therefore, it follows that the deuteron can be modeled as a composite particle comprising six neutrinos, three from its proton and three from its neutron initial constituents.

The proposed 3D geometry of the deuteron structure is shown in Figure 3. It can be viewed as a neutron placed on top of a proton rotated by $60°$, which coincides with the structure of a normal triangular octahedron, a solid body with six vertices, which is one of the five Platonic solids. At the center of this octahedron resides a stagnant positron, originating from the proton [4,5], which combines with a neutron to form the deuteron. The six neutrinos are located at the vertices of the octahedron, and, as shown in Figure 3, they can rotate around seven symmetrically oriented rotational axes.

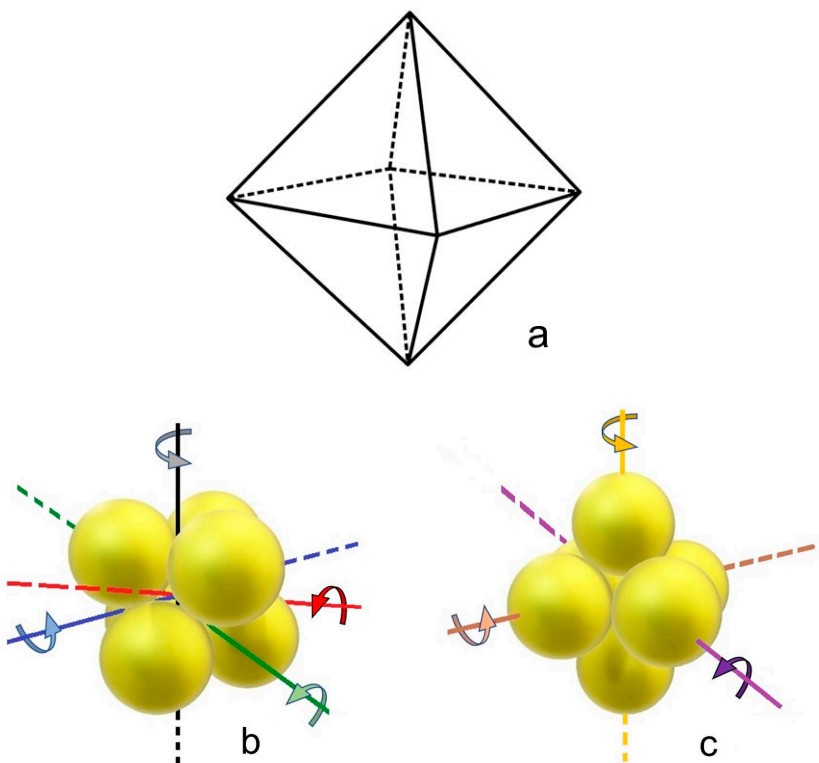

**Figure 3.** Schematic of the RLM structure of the triangular octahedron model (**a**) for the deuteron, drawn using the de Broglie–Compton wavelength spheres of the rotating neutrinos (**b**,**c**) each of radius r = 0.631 fm = $\hbar/\overline{\gamma}m_oc$ and showing the seven rotational axes, of which four are equatorial type (**b**) and three are top–bottom type (**c**).

## 4. Mathematical Modeling and Mass Computation

The deuteron geometry and its mathematical modeling are shown in Figures 3 and 4. The equation of motion of a neutrino in the "equatorial" rotation (Figures 3b and 4a–c) is given in Equation (12).

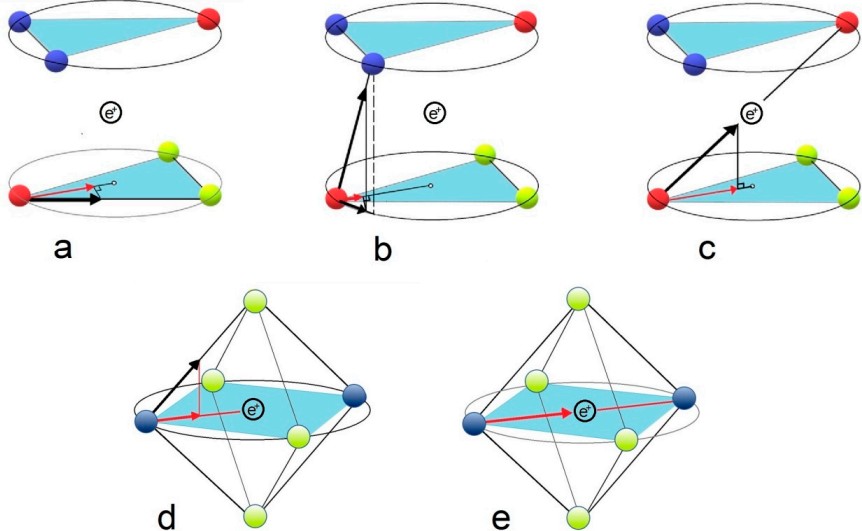

**Figure 4.** Centripetal force components (marked by red arrows) of the forces $F_1$ and $F_2$ for the equatorial (**a**–**c**) and top–bottom (**d**,**e**) neutrino rotations, respectively. Equatorial forces: $F_1 = 2F_a + 2F_b + F_c$; $F_a = (1/2\sqrt{3})(Gm^2/r^2)$; $F_b = (1/6\sqrt{3})(Gm^2/r^2)$; $F_c = (1/3\sqrt{6})(Gm^2/r^2)$. Top–bottom forces: $F_2 = 4F_d + F_e$; $F_d = (\sqrt{2}/6)(Gm^2/r^2)$; $F_e = (1/6)(Gm^2/r^2)$.

$$F_1 = \gamma_1 m_o v_1^2 / r_1 = \left[ G m_o^2 \gamma_1^6 / r^2 \right] \left( 3^{-1/2} + 3^{-3/2} + 2 \times 6^{-3/2} \right), \tag{12}$$

which, together with the de Broglie quantization condition for a composite particle of nuclear spin 1, i.e.,

$$r = r_1 = \frac{2\hbar}{\gamma_1 m_o v_1}, \tag{13}$$

and with the definition of the Planck mass ($m_{Pl} = (\hbar c / G)^{1/2}$), gives for $v_1 \approx c$ the expression

$$\gamma_1 = 2^{1/6} \left( 3^{-1/2} + 3^{-3/2} + 2 \times 6^{-3/2} \right)^{-1/6} m_{Pl}^{1/3} m_o^{-1/3}, \tag{14}$$

Similarly, the equation of motion of a neutrino in the top–bottom rotation (Figures 3c and 4d,e) is

$$F_2 = \gamma_2 m_o v_2^2 / r_2 = \left[ G m_o^2 \gamma_2^6 / r^2 \right] \left( \frac{2\sqrt{2}}{3} + \frac{1}{6} \right), \tag{15}$$

which, together with the de Broglie quantization condition,

$$r_2 = \left( \sqrt{6}/2 \right) r = \frac{2\hbar}{\gamma_2 m_o v_2}, \tag{16}$$

gives

$$\gamma_2 = 2^{1/6} \left( 2^{1/2} + 1/4 \right)^{-1/6} m_{Pl}^{1/3} m_o^{-1/3}. \tag{17}$$

Thus, upon noting that there are four equatorial rotational axes and three top–bottom rotational axes (Figure 3a,b and Figure 4), and accounting for energy equipartition, it follows that the mean rotational $\gamma$ value is given by

$$\overline{\gamma} = (4/7)\gamma_1 + (3/7)\gamma_2. \tag{18}$$

Consequently, accounting for energy conservation, the rotating deuteron mass, $m_{d,r}$, is computed from the composite particle relation:

$$m_{d,r} = 6\overline{\gamma} m_o, \tag{19}$$

or, using Equations (14) and (17)–(19), from

$$m_{d,r} = 2^{1/6} \left[ (4/7) \left[ 3^{-1/2} + 3^{-3/2} + 2 \times 6^{-3/2} \right]^{-1/6} + (3/7) \left[ 2^{1/2} + 1/4 \right]^{-1/6} \right] \left( m_{Pl} m_o^2 \right)^{1/3} \tag{20}$$

To this value must be added the mass of the positron, i.e., 0.511 MeV/c$^2$ [1,11], located at the center of the structure, to obtain

$$m_d = m_{d,r} + m_e = m_{d,r} + 0.511 \text{ MeV/c}^2. \tag{21}$$

Using the $m_o$ value of 0.0437250 eV/c$^2$ found to yield exactly the neutron mass via Equation (9), Equations (19) and (20) give $m_d$ = 1876.368 MeV/c$^2$. Exact agreement with the experimental value of $m_d$ = 1875.613 MeV is obtained for the value $m_{o,d}$ = 0.0436986 eV/c$^2$. This value differs by less than 0.06% from the $m_o$ value, denoted $m_{o,n}$ = 0.0437250 eV/c$^2$, which gives exact agreement with the neutron mass $m_n$ (Equation (9)). Furthermore, if we use $m_{o,n}$ or $m_{o,d}$ to compute both the neutron and the deuteron masses from Equations (9), (19), and (20), the computed values differ by less than 0.05%.

It is worth noting that the rotational radii computed from Equations (13) and (16) are 1.2102 fm and 1.4821 fm, respectively. Using the larger of the two (1.482 fm) and adding

to it the average Compton wavelength, $\hbar/\overline{\gamma}m_o c$, of the neutrino, i.e., 0.6312 fm, in order to obtain the outer radius of the deuteron, one finds $r_d = 2.1133$ fm, which differs by less than 2% from the experimental value of 2.1413 fm.

## 5. Conclusions and Future Work

The astonishing agreement between the computed $m_d$ value (1876.37 MeV/$c^2$) and the experimental value (1875.612 MeV/$c^2$), without use of any adjustable parameters, shows that, similarly to the strong force [4,5,10], the residual strong force or nuclear force can also be modeled quantitatively as relativistic gravity. This force is computed via Equation (5), which accounts for the special relativistic equation for gravitational mass $m_g (= \gamma^3 m_o)$ [2,3] in conjunction with Newton's universal gravitational law.

It appears reasonable to expect that the present RLM approach can be useful for computing interaction energies in larger and more complex nuclei as well, in which case the use of approximate [17,18] internuclear, e.g., Yukawa type, potentials might no longer be necessary, and can be replaced by gradient-enhanced potentials (e.g., gradient modification of Newton's classical gravitational potential). We plan to address this point in future work.

More importantly, the present results imply that both the strong force and the nuclear force can be modeled and computed accurately via the combination of Newtonian gravity and Einstein's special relativity. The high level of agreement between the experimental values and the RLM predictions, both in the case of the strong force in hadrons [4,5,11] and bosons [6,8,11], and in the present modeling description of the nuclear force in the deuteron nucleus, is apparently due to the fact that the RLM combines in a direct way relativity with quantum mechanics via the Einstein and de Broglie equations (Equations (6) and (7), respectively). This combination of relativity and quantum mechanics has been a long-sought goal, expected to lead to a powerful unifying gravitational theory [19–21]. The present results show that this is indeed the case, by demonstrating the unification of gravity with the strong force, including the residual strong force or nuclear force. This, in conjunction with the recently described combination of the weak nuclear force with relativistic gravity [6,11], leaves only two forces, i.e., gravity and electromagnetism.

**Author Contributions:** Conceptualization, C.G.V., E.C.A., D.G., D.T. and K.P.; Methodology, C.G.V., E.C.A., D.G., D.T. and K.P.; Formal analysis, C.G.V., E.C.A., D.G., D.T. and K.P.; Writing-original draft preparation, C.G.V.; Writing-review and editing, C.G.V., E.C.A., D.G., D.T. and K.P.; Supervision, C.G.V. and E.C.A. All authors have read and agreed to the published version of the manuscript.

**Funding:** This research received no external funding.

**Acknowledgments:** We thank our reviewers for their very thorough and constructive reviews.

**Conflicts of Interest:** The authors declare no conflict of interest.

## Glossary/Nomenclature/Abbreviations

| | |
|---|---|
| QCD | Quantum chromodynamics |
| RLM | Rotating lepton model |
| $m_1, m_2, m_3$ | Rest neutrino eigenmasses |
| $m_n$ | Neutron mass |
| $m_e$ | Electron mass |
| $m_\mu$ | Muon mass |
| $m_{Pl}$ | $(\hbar c/G)^{1/2} = 1.221 \times 10^{28}$ eV/$c^2$, Planck mass |
| $\gamma$ | Lorentz factor $= (1 - v^2/c^2)^{-1/2}$ |
| $m_o$ | Rest mass |
| v | Particle speed |
| r | Rotational radius |
| G | Gravitational constant, $6.6743 \times 10^{-11}$ m$^3$/(kgs$^2$) |

| | |
|---|---|
| $m_g$ | Gravitational mass |
| $m_i$ | Inertial mass |
| $c$ | Speed of light in vacuum |
| $\hbar$ | Reduced Planck's constant |
| $eV/c^2$ | Mass unit, $1.783 \times 10^{-36}$ kg |
| $\overline{\gamma}$ | Mean $\gamma$ value |
| $F_1$ | Centripetal force in the equatorial rotation |
| $F_2$ | Centripetal force in the top–bottom rotation |
| $m_d$ | Total deuteron mass |
| $m_{d,r}$ | Mass of the rotating components of the deuteron |
| $m_{o,d}$ | Neutrino rest mass in the deuteron |
| $m_{o,n}$ | Neutrino rest mass in the neutron |
| 1 | Equatorial rotation |
| 2 | Top–bottom rotation |
| d | Deuteron |
| Quantum chromodynamics (QCD) | The strong force theory of the standard model (SM). |
| Rotating lepton model (RLM) | It has the same goals as QCD, but has no adjustable parameters. |
| Deuteron | The nucleus of the deuterium, comprising a proton and a neutron. |
| Quarks | Building blocks of hadrons, identified as relativistic rotating neutrinos in the RLM. |
| Strong force | Force binding the constituents of hadrons. |
| Relativistic gravity | Newtonian gravity, also accounting for special relativity. |
| Residual strong force | The force binding nucleons with other nucleons. |

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
