# Peer review of "Computation of the Deuteron Mass and Force Unification via the Rotating Lepton Model"

_axioms, doi:10.3390/axioms11110657_

Round 1

Reviewer 1 Report

This paper could be considered for publication in the MDPI

Journal of Axioms, if the authors can address some points that require attention. Authors should consider the following remarks:

  1. Authors should state clearly what they have done by giving the details of the obtained results and specifying the main novelty of this article. 
  2. Authors are advised to create a section for the definitions of some terminologies used in this paper.
  3. Authors should check all the mathematical equations errors and retype them correctly. Authors are advised to use Latex if possible.
  4. Authors should include a conclusion section and suggest future research work. 
  5. Authors should review the whole manuscript, correct grammatical and typo errors, and improve their English.

Author Response

The main amendments are the following:

1.  Three words were added to the title for additional clarity.

2.  The Abstract was also modified to further elucidate the computational procedure.

3.  Section 2, describing the Rotating Lepton Model (RLM) was modified for added clarity regarding the combination of special relativity with Newtonian gravity in equation (5).

4.  A “Conclusions” section was added in order to underline and emphasize the novelty and importance of the present work, as suggested.

5.  A list of definitions of terminologies and a list of symbols has been also added as suggested by our reviewers.

The above changes have been made using “Track changes” in MS Word.

We are thankful to our reviewers for their valuable comments. This is now mentioned in the Acknowledgement.

Reviewer 2 Report

This paper extends the 2D rotating lepton model to 3D, and uses the model to compute the mass of the Deuteron. The computation does not rely on adjustable parameters, and yields a value that agrees well with the experimental one. Some conclusions are obtained.

The paper is very interesting and the findings seem sound.

I suggest:

to include a section for conclusions;

to include a list of symbols;

Author Response

(The authors gave the same response as above.)

Reviewer 3 Report

attached

Author Response

(The authors gave the same response as above.)

Round 2

Reviewer 1 Report

The quality of the paper has been improved by addressing most reviewers' comments.  Authors are advised to recheck the whole paper and correct indentation, mathematical equations, typos, and grammatical errors.